# Air-Filled Bubbles Stabilized by Gold Nanoparticle/Photodynamic Dye Hybrid Structures for Theranostics

**DOI:** 10.3390/nano11020415

**Published:** 2021-02-06

**Authors:** Roman A. Barmin, Polina G. Rudakovskaya, Olga I. Gusliakova, Olga A. Sindeeva, Ekaterina S. Prikhozhdenko, Elizaveta A. Maksimova, Ekaterina N. Obukhova, Vasiliy S. Chernyshev, Boris N. Khlebtsov, Alexander A. Solovev, Gleb B. Sukhorukov, Dmitry A. Gorin

**Affiliations:** 1Skolkovo Institute of Science and Technology, 3 Nobelya Str., 121205 Moscow, Russia; Roman.Barmin@Skoltech.ru (R.A.B.); P.Rudakovskaya@skoltech.ru (P.G.R.); O.Sindeeva@skoltech.ru (O.A.S.); Elizaveta.Maksimova@skoltech.ru (E.A.M.); E.Obukhova@skoltech.ru (E.N.O.); V.Chernyshev@skoltech.ru (V.S.C.); g.sukhorukov@skoltech.ru (G.B.S.); 2Remote Controlled Theranostic Systems Lab, Saratov State University, 83 Astrakhanskaya Str., 410012 Saratov, Russia; olga.gusliakova17@gmail.com (O.I.G.); prikhozhdenkoes@gmail.com (E.S.P.); 3Institute of Biochemistry and Physiology of Plants and Microorganisms, Russian Academy of Sciences, 13 Prospekt Entuziastov, 410049 Saratov, Russia; khlebtsov_b@ibppm.ru; 4Department of Materials Science, Fudan University, Shanghai 200433, China; solovevlab@gmail.com; 5School of Engineering and Materials Science, Queen Mary University of London, Mile End Rd, London E1 4NS, UK

**Keywords:** microbubbles, albumin, gold nanoparticles, photodynamic dyes, photoacoustics, fluorescence imaging, ultrasound, zinc phthalocyanine, indocyanine green

## Abstract

Microbubbles have already reached clinical practice as ultrasound contrast agents for angiography. However, modification of the bubbles’ shell is needed to produce probes for ultrasound and multimodal (fluorescence/photoacoustic) imaging methods in combination with theranostics (diagnostics and therapeutics). In the present work, hybrid structures based on microbubbles with an air core and a shell composed of bovine serum albumin, albumin-coated gold nanoparticles, and clinically available photodynamic dyes (zinc phthalocyanine, indocyanine green) were shown to achieve multimodal imaging for potential applications in photodynamic therapy. Microbubbles with an average size of 1.5 ± 0.3 μm and concentration up to 1.2 × 10^9^ microbubbles/mL were obtained and characterized. The introduction of the dye into the system reduced the solution’s surface tension, leading to an increase in the concentration and stability of bubbles. The combination of gold nanoparticles and photodynamic dyes’ influence on the fluorescent signal and probes’ stability is described. The potential use of the obtained probes in biomedical applications was evaluated using fluorescence tomography, raster-scanning optoacoustic microscopy and ultrasound response measurements using a medical ultrasound device at the frequency of 33 MHz. The results demonstrate the impact of microbubbles’ stabilization using gold nanoparticle/photodynamic dye hybrid structures to achieve probe applications in theranostics.

## 1. Introduction

Microbubbles filled with air and various gases have been widely studied over the past decades. Air microbubbles are widely used in medicine [1], food chemistry [2], technology [3], as micromotors [4], and other fields [5]. The chemical composition of microbubbles can be different, and the structure of the shell varies from lipids, polymers, proteins to small bifunctional molecules [6,7,8,9]. The bubbles’ shell structure and their stability are important factors to consider for their potential uses, which depends on the core-shell engineering methods [10,11].

One of the most interesting agents for producing vesicles is protein [2,8,12,13], particularly bovine serum albumin (BSA), which is used in this work. The chemical nature of these microbubbles and the origin of their extremely long life span have been investigated. The chemical mechanism by which high-intensity ultrasound creates air-filled protein microbubbles from aqueous protein solutions is described in [14]. The microbubbles are held together mainly by protein–protein crosslinking of cysteine residues. The main cross-linking agent is superoxide, which forms as a result of the extremely high temperatures generated during acoustic cavitation [15]. However, another study [16] showed that microbubbles can be produced with proteins lacking cysteine residues. The formation and stability of microbubbles are due to hydrophobic interactions. In 2008, lysozyme-based microbubbles with additional available thiol groups were obtained [15], in which the binding was carried out both due to hydrophobic interactions and due to crosslinking with disulfide bridges.

Biomedical microbubbles are mainly used as ultrasound (US) contrast agents [17,18,19,20]. Another application of microbubbles is for intravenous oxygen delivery [21,22,23]. To ensure unimpeded movement through the circulatory system, such microbubbles have an optimum diameter of less than 10 μm. An interesting research topic involves the addition of a second functionality—magnetic resonance imaging (MRI) contrast, dye or a photodynamic therapy (PDT) agent. The achievement of microbubble stability is required for a wide range of biomedical applications, which can be provided by proteins and lipids whereas the use of a polymer as a shell material can create an overly rigid shell, which can reduce the usefulness of microbubbles as ultrasonic contrast agents [24].

To perform bimodal tasks such as simultaneous US and photoacoustic (PA) imaging, i.e., the combination of ultrasound imaging with MRI and CT, additional modules must be introduced into the vesicle structure. For instance, in articles [25,26], gold nanorods were obtained and introduced into the structure of bubbles as a double PA/US contrast agent. Subsequently, the application can be extended for diagnostic purposes if gold surface can be modified with targeted functional ligands [27]. The hybrid method can simultaneously display anatomical and functional information, and show images with fair spatiotemporal resolution. Besides, various contrast agents’ physical characteristics can expand the field of application to a therapeutic effect. It is known that gold nanorods have tunable plasmon resonance and other remarkable optical properties [28,29]. In another study, the authors of [30] prepared smart gold nanoparticle-stabilized microbubbles composed of a gas-filled core and a shell including smart gold nanoparticles (SAuNPs), which can be aggregated in tumors and used in ultrasound-mediated cancer theranostics. Microbubbles aggregated in tumors allow photoacoustic monitoring and photothermal treatment of tumors.

Photodynamic therapy is now well established for the treatment of cancer, and multiple autoimmune and infectious diseases [31,32]. Chemical compounds that are used as photosensitizers can be divided into several groups, including non-porphyrin compounds, the most developed drugs based on cyanines (for example, indocyanine green (ICG)), as well as phthalocyanines, which are aromatic heterocycles consisting of four isoindole rings connected by nitrogen atoms and capable of coordinating metal in the center of the molecule (Photosens, Holosens) [33,34]. The main advantages of these compounds are chemical homogeneity, absorption of light in the long-wavelength range of the spectrum (675–700 nm), high extinction coefficient, and high quantum yield of singlet oxygen. The possibility of chemical modification is an essential chemical feature of these photosensitizers. In this work, these molecules were used for cross-linking using a protein molecule that makes it possible to quantitatively introduce a photosensitizer-dye into the structure of bovine serum albumin, and then into microbubbles. This is a new topic that has not yet been studied in the literature. The combination of PDT agents with microbubbles opens up many possibilities for multifunctional diagnostics and therapy. It will also allow the use of sonodynamic therapy [35], which combines ultrasound, oxygen and the use of photosensitizers as an adjunct to PDT. The literature [36,37,38] mentions the use of indocyanine green for sonodynamic therapy, but without bubbles.

Many parameters influence the number, size, stability and other physicochemical characteristics of microbubbles including initial protein concentration, pH, ionic strength of the solution, the temperature of the initial solution and synthesis, time and intensity of sonication, and storage conditions [13,15,39,40,41]. It was shown that increasing the protein concentration from 0.5% (w/v) to 5.0% (w/v) resulted in an increase in the yield and stability of microbubbles, and affected the size. In the literature, the effect of preheating and ionic strength on the yield and stability of microbubbles is only partially described. In [16], the authors found that microbubbles are less stable at higher temperatures while storage in a refrigerator at 4 °C prolongs bubbles’ life and stability. Results [2,13] have shown that the bubbles are stable when stored at 5 °C and that the size of microbubbles decreases during storage. This is because the microbubble shell is less rigid at high temperatures. The above studies provide some insight into the influence of the preparation parameters on the quality and quantity of microbubbles.

In this work, it will be shown that a number of parameters affect the bubble size. The number of microbubbles obtained, their size and their stability over time were studied depending on the protein concentration, ionic strength (saline solutions were used), the significance of the preheating temperature and storage temperature. Also, we report on experimental studies conducted with regard to the synthesis, storage stability and in vitro stability of bovine serum albumin (BSA)-coated microbubbles containing air core. These microbubbles were obtained by sonication using formulations containing BSA and additives that provide the resulting bubbles’ functional characteristics. As additives, we used gold nanoparticles coated with bovine serum albumin (AuNPs), as well as dyes cross-linked with a protein using photodynamic activity—zinc phthalocyanine (ZnPc) and ICG. These additives were used in various combinations, resulting in 6 basic samples: bubbles based on pure bovine serum albumin, bubbles containing ZnPc, bubbles containing ICG, bubbles containing gold, and complex structures containing gold and ZnPc, gold and ICG. Freshly prepared polydisperse samples were purified by dialysis against a saline solution. In this work, we studied the stability of the samples over time, bubbles physicochemical characteristics, and their potential uses in the biomedical field. Spectroscopic analyses were performed to study the effects of dye and gold additions. It was found that the addition of gold to the composition, even at low concentrations, increased the stability of the samples and the mono-dispersity of the resulting microbubbles. Microbubbles were obtained and tested in saline, which enables further in vivo studies to be carried out. In addition to this, in vitro US studies performed using modified microbubbles have also shown promising results.

## 2. Materials and Methods

### 2.1. Materials

Bovine serum albumin (BSA), Chloroauric acid (HAuCl4*3H2O), Sodium citrate monobasic (HOC(COONa)(CH2COOH)2), N-(3-Dimethylaminopropyl)-N′-ethylcarbodiimide hydrochloride (EDC), N-Hydroxysuccinimide (NHS), Sodium chloride (NaCl), and agarose were all purchased from Sigma-Aldrich (Darmstadt, Germany). Holosens^®^, octachloride octakis [*N*(2-hydroxyethyl)-*N*,*N*,-(dimethylammoniomethyl)] zinc (II) phthalocyanine (ZnPc) was chosen as a photodynamic dye [33,42] and purchased from the Organic Intermediates and Dyes Institute (Moscow, Russia). Indocyanine green was also used as a photodynamic dye because of its long-time clinical application and approval for use by the U.S. Food and Drug Administration [43] and was purchased from Dynamic Diagnostics (Plymouth, MI, USA). Deionized (DI) water with specific resistivity, higher than 18.2 MΩm from a Milli-Q Integral 3 water purification system (Millipore, Burlington, MA, USA), was used to make all solutions.

### 2.2. Methods

#### 2.2.1. AuNPs Synthesis

The synthesis was carried out according to the modified synthesis of Turkevich [44]. A solution of 75 mg gold tetrachloroaurate in 110 mL of distilled water was placed in a 250 mL round-bottom flask equipped with a reflux condenser and the solution was brought to a boil. Then, 26.25 mL of 1% sodium citrate solution was quickly added and boiling was continued for one hour. On boiling, the solution color changed from light yellow to cherry through dark tones. With stirring, the solution was cooled to room temperature. Then the particles were used in the form of a colloidally stable solution.

The modification of the surface of gold nanoparticles was carried out according to the technique developed by our group. A similar technique was proposed by the authors [45] for replacing cetyltriethylammonium bromide with bovine serum albumin. In this work, citrate, which stabilizes the surface of nanoparticles, was replaced by protein. A solution of gold nanoparticles (concentration 130 ± 4 mg/L, 10^14^ particles/L) with a volume of 10 mL was slowly poured into a solution containing bovine serum albumin with a concentration of 10 mg/mL, sodium citrate 0.1% and having pH = 12 in a ratio of 1:1 by volume. The resulting solution was kept in an ultrasonic bath for 30 min. Then the nanoparticles were centrifuged, washed with a protein solution (10 mg/mL) and re-suspended in 5 mL BSA solution (1 mg/mL, pH = 7). Zeta-potential and dynamic light scattering measurements for AuNPs before and after BSA coating are provided in Appendix A.

#### 2.2.2. BSA-ZnPc Preparation

The synthesis of a complex of BSA with zinc phthalocyanine (ZnPc) was carried out by carbodiimide synthesis. 30 mg of bovine serum albumin was dissolved in 7 mL of phosphate buffer (pH = 8), 775 μL of EDC (solution in PBS with a concentration of 1 mg/mL) was added to the resulting solution, stirred for 15 min, then a solution of N-hydroxysuccinimide 1.05 mL (solution in PBS 1 mg/mL) was added, mixed for another 15 min, and 1 mL of Holosens (1 mg/mL) was added. The solution was stirred for 12 h in the cold (4 °C). The solution was then washed by dialysis against water.

#### 2.2.3. BSA-ICG Preparation 

Bovine serum albumin (100 mg) was dissolved in 6 mL of phosphate buffer (pH = 7.4) and mixed with 1 mL of indocyanine green (ICG) solution with a concentration of 7 mg/mL, then the mixture was stirred for 3 h. The solution was then washed by dialysis against water for 48 h in the cold (4 °C). After dialysis the solution was diluted twice.

#### 2.2.4. Mass Spectrometry Measurements

Samples containing BSA were analyzed using a time-of-flight mass spectrometer with matrix laser desorption/ionization (MALDI-TOF/TOF) rapifleX MALDITOF/TOF MS System (Bruker Daltonik GmbH, Bremen, Germany). The operating mode was as follows: linear mode, positive ionization, analysis range m/z 5000–70,000, accelerating voltage 20 kV, SmartBeam III laser, laser frequency 10 kHz, frequency 200 Hz. Before analysis, the device was calibrated using a mixture of proteins, “Protein Calibration Standard I” (Bruker Daltonik GmbH, Bremen, Germany). The mixture included the following proteins: insulin ([M + H] = m/z 5734.5), ubiquitin I ([M + H] = m/z 8565.76), cytochrome C ([M + H] = m/z 12,361.2), myoglobin ([M + H] = m/z 16,952.5). 2.5-dihydroxybenzoic acid (Bruker Daltonik GmbH, Bremen, Germany) with purity > 99.0% was used as the matrix. A 20 mg/mL matrix solution was prepared in a mixture of 30% acetonitrile:70% water:0.1% trifluoroacetic acid. An aqueous solution of the samples was mixed with the matrix in a ratio of 1:1 and 1 μL of the mixture was applied to the target plate.

The ZnPc sample was analyzed using a time-of-flight mass spectrometer with matrix laser desorption/ionization (MALDI-TOF/TOF) rapifleX MALDITOF/TOF MS System (Bruker Daltonik GmbH, Bremen, Germany). The operating mode was: reflector mode, positive ionization, analysis range m/z 300–2000, accelerating voltage 20 kV, SmartBeam III laser, laser frequency 10 kHz, frequency 200 Hz. Before analysis, the device was calibrated using a mixture of peptides, “Peptide Calibration Standard II” (Bruker Daltonik GmbH, Bremen, Germany). The mixture includes peptides with a mass range of 700–3200 Da. 2.5-dihydroxybenzoic acid (Bruker Daltonik GmbH, Bremen, Germany) with purity > 99.0% was used as the matrix. A 20 mg/mL matrix solution was prepared in a mixture of 30% acetonitrile:70% water:0.1% trifluoroacetic acid. An aqueous solution of the sample was mixed with the matrix in a ratio of 1:1 and 1 μL of the mixture was applied to the target plate.

#### 2.2.5. Surface Tension Measurements

Custom software implemented in Matlab (Mathworks, Natick, MA, USA) and described in detail previously, was used to capture and process pendant drop images for surfactant characterization [46,47]. The actual update rate of the surface tension measurements, droplet volume, and surface area was 2 s, constrained by the time needed to perform the necessary calculations on a particular computer used by us (Dell Latitude 7280). At the preprocessing step, the pendant droplet boundary was determined by converting a grayscale image into a binary image by using the threshold value calculated by Otsu’s method [48]. For each row of pixels, a midpoint between boundary points of the droplet was determined. The vertical centerline position that divides the droplet into two symmetrical halves was then obtained by averaging midpoint values obtained for each row of pixels. The surface tension of the drop was then found by solving the Young-Laplace equation at each time point by using the system identification theory that minimizes the difference between the theoretically predicted and the imaged shape of the interface [49]. For each measurement, a pendant drop was formed, and surface tension was obtained in real-time for 10 min at room temperature using the described Matlab software v. R2019a (Mathworks, Natick, MA, USA). The measurement was repeated 4 times. The final result represents the average surface tension and standard deviation of the last 2 min of the 4 repeated measurements.

#### 2.2.6. Microbubbles Preparation

Microbubbles were obtained by the modified sonication method [2,14,50]. Briefly, for each sample, 150 mg of BSA were dissolved in 1 mL of 2.7% NaCl aqueous solution, in order to produce each microbubbles sample from the components dissolved in saline solution. For samples labeled with ZnPc, BSA-ZnPc solution (1 mL) was added in each sample; then, for the sample containing AuNPs (BSA-ZnPc-AuNPs MBs), AuNPs solution (1 mL) was added, while for sample labeled with ZnPc only (BSA-ZnPc MBs), an aqueous solution (1 mL) was added. Likewise, for samples labeled with ICG, BSA-ICG solution (1 mL) was added in each sample; then, AuNPs solution (1 mL) was added for the sample containing BSA coated AuNPs (BSA-ICG-AuNPs MBs), while for sample labeled with ICG only (BSA-ICG MBs), only an aqueous solution (1 mL) was added. For the preparation of bubbles with the BSA shell without any additional shell modification, 2 mL of DI water were added to 1 mL of BSA solution obtained as described above. All samples were stored in a glass vial and heated to a temperature of 50 °C to lower the solution’s surface tension. Each sample was sonicated for 5 min at the maximum power of 100 W on the Bandelin Sonopuls HD4100 sonicator with the TS103 sonotrode probe (Bandelin Electronic GmbH & Co KG, Lueneburg, Germany). The tip of the sonotrode was placed at the interface between the phases of liquid solution and air. After sonication, each sample was stored at 4 °C for 30 min for further stabilization. Then, all produced samples with microbubbles were dialyzed at 4 °C for 12 h in saline solution.

#### 2.2.7. Optical Microscopy

Optical microscopy (OM) was carried out on an Olympus CX33 (Olympus Corporation, Tokyo, Japan). The size distribution of microbubbles was evaluated using images of 200 microbubbles.

#### 2.2.8. Microbubbles Concentration Measurements

Microbubbles concentration was determined with the use of the Gorjaev’s chamber: briefly, 10 μL solution with microbubbles after dialysis (without dilution and with 5- or 10- times dilution) were injected between the glass slides, then stored at room temperature for 5 min in order for the bubbles to float to the upper glass slide. Then, photographs were taken with the optical microscope within the grid of the chamber. For each sample, more than 200 microbubbles were counted to determine the concentration of the sample. For each bubbles probe, concentrations were determined 30 min after the sonication and (0 h), after storage at 4 °C for 12 h before and after dialysis, and after storage at 4 °C for 36 h before and after dialysis. Each measurement was repeated 5 times.

#### 2.2.9. Transition Electron Microscopy

Transmission electron microscopy (TEM) images were obtained on a Titan Themis Z (TFS (ThermoFisherScientific), Breda, The Netherlands)—gold nanoparticles, Zeiss M912 Omega transmission electron microscope (Carl Zeiss Microscopy GmbH, Jena, Germany)—MBs at an operating voltage of 300 kV.

#### 2.2.10. Zeta-Potential Measurements

Zeta-potential measurements were performed on the ZetaSizer Nano ZS analyzer (Malvern Panalytical, Malvern, UK); all measurements were diluted 20 times in DI water and placed in a U-cuvette, carried out at 25 °C and repeated three times.

#### 2.2.11. Extinction Spectra Measurements

Extinction spectra were measured using a multifunctional microplate reader Tecan Infinite M Nano+ (Tecan Trading AG, Männedorf, Switzerland) at room temperature (25 °C), where samples were placed in a plastic 96-well plate. All samples were diluted in saline with concentrations of 1 × 10^8^, 5 × 10^7^, 2.5 × 10^7^, 1.25 × 10^7^, 6.25 × 10^6^ bubbles/mL.

#### 2.2.12. Fluorescence Tomography Measurements

For fluorescence tomography measurements, each sample was diluted in saline and added in a 96 well plate in the same manner as for extinction spectra measurements. The plate with samples was then imaged by the IVIS CT Spectrum In Vivo system (Xenogen Corp., San Francisco, CA, USA) at room temperature (25 °C). Sequence images were acquired with the Excitation/Emission pair of 675/720 nm for samples containing ZnPc and the pair of 745/840 nm for samples containing ICG. Exposure time is auto, FOV = C. Photons were quantified with the LivingImage software v.4.5.3 (Xenogen Corp., Alameda, CA, USA).

#### 2.2.13. Raster-Scanning Optoacoustic Mesoscopy Measurements

A raster-scanning optoacoustic mesoscopy system (RSOM) Explorer P50 (iTheraMedical GmbH, Munich, Germany) was used to collect optoacoustic signals from samples. The optoacoustic signals were collected by a custom-made, spherically focused LiNbO3 detector (center frequency—50 MHz; bandwidth—11–99 MHz; focal diameter—3 mm; focal distance—3 mm). The samples were irradiated by a frequency-doubled flashlamp-pumped Nd:YAG laser (wavelength—532 nm, pulse duration—2.5 ns; pulse energy—200 μJ; repetition rate—1 kHz). The repletion rate of 1 kHz was used from the options of 0.5, 1 or 2 kHz. Light from the laser is delivered through a glass fiber 2-arm bundle (spot size—3.5–5 mm). The scanning head is mounted to two motorized stages (field view up to 12 × 12 × 4 mm^3^). The samples were tested in the agarose phantom. For the preparation of a phantom, agarose (100 mg) was diluted in DI water (10 mL) at room temperature, then the solution was stirred intensively at a temperature of 100 °C, and then degassed to avoid the presence of small air bubbles in the solution. Briefly, for a phantom formation, a droplet of agarose (30 μL) was placed on the bottom of the reservoir, and then after 15 s, a droplet of a sample (7 μL) was injected into the upper third of the formed agarose droplet, forming a liquid reservoir of the sample inside the phantom. Additional storage at fridge conditions (4 °C) was applied for 15 min to solidify the phantom, and then the phantom with the sample was covered with a layer of DI water (1.5 cm) to carry out the measurements. The scan head was coupled to the sample by a water-filled reservoir, and the samples were scanned over the field of view (8 × 8 mm^2^) with a predefined depth (4 mm). 

#### 2.2.14. Ultrasound Characterization

The DUB^®^ Skinscanner (Taberna Pro Medicum GmbH, Lueneburg, Germany) was used to evaluate ultrasound contrast of obtained bubbles with a 33 MHz applicator (depth of scanning 8 mm, axial resolution 42 μm). The received signals from the applicator were processed using DUB SkinScanner software v.5.31 (Taberna Pro Medicum GmbH, Lueneburg, Germany).

## 3. Results and Discussion

### 3.1. Optimization of the Experiment and Microbubbles’ Stabilization with Hybrid Structures

The modified sonication method was used to produce the bubbles, and the following conditions were examined: the power amplitude of the sonotrode, time of sonication procedure, and temperature were varied. The most suitable conditions for preparing stable probes were the following: power of 100 W, sonication for 5 min, and the sonotrode tip’s location at the surface of a solution. Additionally, the temperature of the solution was raised to 50 °C to reduce the surface tension of the initial solution, which caused the reduction in the mean size of the produced microbubbles. It should be noted that to complete the formation of bubbles and to obtain stable microbubbles after ultrasonic treatment, the bubbles were placed in a refrigerator for 30 min (4 °C). Therefore, all samples were prepared using the method presented in Figure 1. After the preparation, all bubble-containing samples were stored in fridge conditions for further stabilization and dialyzed for 12 h to purify the sample.

The gaseous core of obtained air-filled microbubbles provides excellent acoustic properties and the potential to serve as an ultrasound (US) contrast agent. [51].

For the implementation of fluorescent (FL) imaging modality and the possibilities of photodynamic therapy approach implementation, two clinically available photodynamic dyes were chosen—zinc phthalocyanine (ZnPc) and indocyanine green (ICG)—and covalently bound to BSA for conjugation of BSA and ZnPc, BSA and ICG, respectively. Such binding was confirmed by mass spectrometry measurements, as one can see in Figure 2.

About one molecule of ZnPc/ICG was bound to one molecule of albumin on each albumin-dye conjugate. As shown in Figure 2c, the curve shape of or the BSA-ZnPc conjugate in the range of 67–70 kDa demonstrated the same behavior as the curve shape observed for the original dye spectrum (presented in Appendix A). 

For the implementation of photoacoustic (PA) imaging and shell stabilization of bubbles, gold nanoparticles (AuNPs) were chosen as functional additives. The results of TEM provided in Figure 3a proved the shape of the nanoparticles and the presence of a protein shell of ~2 nm, which were used for microbubble stabilization.

Additionally, TEM images were taken for dried samples of AuNPs-containing probes: BSA-ZnPc-AuNPs MBs (Figure 3b) and BSA-ICG-AuNPs MBs (Figure 3c). For both samples containing dried bubbles, the presence of AuNPs was observed, and ~50–60 particles were placed on each bubble. The presence of AuNPs was indirectly confirmed during the extinction spectra measurements of bubbles after dialysis (Appendix A): a slight increase in absorption was observed at a wavelength of 520 nm for the BSA-AuNPs MBs and BSA-ZnPc-AuNPs MBs samples, but was less significantly observed for the BSA-ICG-AuNPs MBs sample due to the significant influence of the dye (ICG) extinction used in the system. However, for all AuNPs-containing samples, a broadening of the nanoparticles’ characteristic peak was observed due to the protein envelope: BSA’s influence was observed.

### 3.2. Samples Characterization: Concentration and Mean Size of Stabilized Microbubbles

The concentration for each probes’ sample was evaluated with Gorjaev’s chamber using optical microscopy (OM) images. The concentration of the bubbles are presented in Figure 4. The sample concentration was evaluated for probes without dialysis during storage time at 1, 12, and 36 h after preparation, and for dialyzed probes at 12 and 36 h after preparation. All samples were stored at similar fridge conditions to measure the stability of probes.

In ascending order of the number of bubbles immediately after preparation, the samples can be arranged as follows: BSA MBs (with concentration of 4.4 × 10^8^ MBs/mL), BSA-AuNPs MBs (5.4 × 10^8^ MBs/mL), BSA-ZnPc MBs (8.5 × 10^8^ MBs/mL), BSA-ZnPc-AuNPs MBs (8.9 × 10^8^ MBs/mL), BSA-ICG MBs (1.1 × 10^9^ MBs/mL), BSA-ICG-AuNPs MBs (1.2 × 10^9^ MBs/mL). The dye’s inclusion in the bubble shell increased the initial concentration of bubbles during preparation compared to the shell consisting of albumin only, and ICG had the greatest influence compared with ZnPc. Additionally, the stabilization of the probes’ shell with AuNPs increased the concentration after preparation and optimized the stability properties for the obtained probes, which is especially apparent in a pairwise comparison of BSA MBs: BSA-AuNPs MBs, BSA-ZnPc MBs, BSA-ZnPc-AuNPs MBs, and BSA-ICG MBs and BSA-ICG-AuNPs MBs samples.

Dialysis resulted in bubbles-containing samples free from impurities of components that are not bound in the shell structure; however, mechanical effects introduced during the procedure may affect the samples’ stability. After 12 h of storage, the bubbles containing AuNPs only (BSA-AuNPs MBs with a concentration of 3.6 × 10^8^ MBs/mL) were more stable than the ZnPc-containing bubbles (BSA-ZnPc MBs with a concentration of 1.4 × 10^8^ MBs/mL, and BSA-ZnPc MBs with a concentration of 3.0 × 10^8^ MBs/mL), while the samples containing ICG were the most stable. The BSA-ICG MBs had the most excellent stability with a concentration of 7.0 × 10^8^ MBs/mL, and BSA-ICG-AuNPs MBs demonstrated a concentration of 5.4 × 10^8^ MBs/mL. OM images for samples with and without dialysis 12 h after preparation demonstrated such behavior as the addition of stabilization and functionalization components influence on concentration, as presented in Figure 5.

The zeta-potential of all the bubbles stabilized with nanoparticles- and/or dyes- hybrid structures revealed good stability with values of −5.3 ± 0.9 mV for BSA-ZnPc-AuNPs MBs and −4.8 ± 0.2 mV for BSA-ICG-AuNPs MBs samples, while the probe with BSA-only shell revealed moderate stability with a value of −8.4 ± 0.6 mV for BSA MBs sample, as presented in Appendix A. Surface tension measurements coincide with microbubble stability as BSA-only shell showed the highest surface tension of 52.1 ± 0.4 mN/m while BSA-ZnPc, BSA-ZnPc-AuNPs, BSA-ICG and BSA-ICG-AuNPs showed 50.5 ± 0.4, 50.8 ± 0.4, 51.1 ± 0.3 and 50.9 ± 0.4 mN/m, respectively (average ± standard error of the last 2 min of the dynamic surface tension measurement). Throughout the 10 min surface tension measurement, the average difference between BSA-only and BSA-ZnPc, BSA-ZnPc-AuNPs, BSA-ICG, BSA-ICG-AuNPs surface tension was 1.5 ± 0.3, 1.1 ± 0.3, 1.2 ± 0.3 and 1.5 ± 0.3, respectively. The introduction of dye (ICG/ZnPc) and gold (AuNPs) reduced the surface tension of the initial solution used for the probes’ preparation, which can lead to an increase in the concentration and stability of the obtained samples.

The average size measurements are shown in Figure 6. The measurements were made for all samples treated with or without dialysis at the same storage conditions (4 °C) and at a time after bubble preparation (12 h). As one can see, all bubbles have a mean size of 1–2 μm with a small size dispersion, and the use of dialysis led to a slight decrease (~0.1–0.2 μm) in the average size of the samples. In descending order, samples without dialysis can be arranged as follows: BSA-ICG MBs (1.473 ± 0.482 μm), BSA MBs (1.465 ± 0.276 μm), BSA-ICG-AuNPs MBs (1.464 ± 0.536 μm), BSA-ZnPc-AuNPs MBs (1.416 ± 0.322 μm), BSA-AuNPs MBs (1.403 ± 0.278 μm), and BSA-ZnPc MBs (1.291 ± 0.229 μm). Samples after dialysis can be arranged as: BSA-AuNPs MBs (1.344 ± 0.281 μm), BSA MBs (1.264 ± 0.234 μm), BSA-ICG MBs (1.262 ± 0.273 μm), BSA-ZnPc-AuNPs MBs (1.233 ± 0.192 μm), BSA-ZnPc MBs (1.219 ± 0.239 μm), BSA-ICG-AuNPs MBs (1.205 ± 0.265 μm).

All the described samples had a mean size corresponding to the criteria for contrast agents (they were less than 10 μm) [52,53], and a relatively small size dispersion for bubbles prepared by the sonication method, which confirmed the selected conditions for agents’ preparation and functionalization. Thus, characterization of imaging properties is needed to determine the potential biomedical applications for such probes.

### 3.3. Characterization of Fluorescent, Photoacoustic and Ultrasound Imaging Properties of Microbubbles

Since multimodal imaging can be achieved as a combination of fluorescent (FL), optoacoustic (OA), and acoustic/ultrasound (US) imaging modalities, potential applications of obtained microbubbles in imaging were tested.

The results of the fluorescence tomography measurements are presented in Figure 7. All microbubbles modified with photodynamic dyes (ZnPc and ICG) revealed significant fluorescence intensity at relevant excitation/emission pairs (675/720 nm for ZnPc and 745/840 nm for ICG), while microbubbles containing only BSA shell or shell modified with AuNPs only did not demonstrate comparable fluorescence signals.

The combination of gold nanoparticles and photodynamic dyes’ influence on the fluorescent signal can be seen in Figure 7a,b. The implementation of AuNPs for the bubble shell modified with dyes (BSA-ZnPc-AuNPs MBs and BSA-ICG-AuNPs MBs) led to a decrease in the fluorescent signal compared with probes containing dye only (BSA-ZnPc MBs and BSA-ICG MBs), and fluorescence quenching was observed. The fluorescence total radiant efficiency for BSA-PcZn MBs and BSA-ICG MBs was 1.8 times higher than for BSA-PcZn-AuNPs and MBs BSA-PcZn-AuNPs MBs, respectively, still, the introduction of AuNPs led to an increase in stability of probes and led to the path for potential photoacoustic imaging. The highest fluorescence total radiant efficiency signal was observed for microbubbles containing ICG only. BSA-ICG MBs revealed a 7.8 higher fluorescence signal than BSA-PcZn MBs, and the same phenomena were observed in a comparison of nanoparticles-containing probes (BSA-ICG-AuNPs MBs and BSA-ZnPc-AuNPs MBs pair). As one can see in Appendix A, similar behavior can be observed regarding the dependence of fluorescence on concentration for the obtained microbubbles. At the wavelength of 700 nm for ZnPc, the fluorescence signal of BSA-ZnPc MBs was 2.3 higher than that of BSA-ZnPc MBs; the fluorescence signal of BSA-ICG MBs was 1.7 higher than that of BSA-ZnPc MBs; BSA-ICG MBs demonstrated 3.3-times higher fluorescence than BSA-PcZn MBs, and the fluorescence of BSA-ICG-AuNPs MBs was 4.6 times higher than that of BSA-ZnPc-AuNPs MBs.

Thus, FL imaging modality can be successfully achieved with all dye-containing probes, which can be placed in descending order according to their fluorescence intensity: BSA-ICG MBs, BSA-ICG-AuNPs MBs, BSA-ZnPc MBs, BSA-ZnPc-AuNPs MBs.

Next, probes were tested for their potential application in PA imaging. Since AuNPs are well-known probes for such imaging modalities [54,55,56], such particles were used for bubble shell stabilization and PA imaging implementation. Thus, AuNPs-containing MBs (BSA-AuNPs MBs, BSA-ZnPc-AuNPs MBs, BSA-ICG-AuNPs MBs) and related solutions (AuNPs, BSA-ZnPc, and AuNPs, BSA-ICG and AuNPs in the same concentrations of components used for probes preparation, respectively) were measured by raster-scanning optoacoustic mesoscopy (RSOM), as one can see in Figure 8.

The presence of microbubbles in BSA-AuNPs MBs, BSA-ZnPc-AuNPs MBs, and BSA-ICG-AuNPs MBs can be seen by its localization in the upper part of samples due to bubbles floating, as can be seen in X-Z projections on Figure 8b,d,f, respectively. Additionally, the occurrence of a high-frequency PA signal in the range with a frequency of 33 MHz can be seen for all bubbles-containing samples. It can be seen in the comparison of ZnPc-containing samples (Figure 8c,d) where the specific signal for microbubbles demonstrated a higher specific PA signal compared to the signal shown by gold particles.

Comparing the PA signal obtained for AuNPs saline solution and BSA-AuNPs MBs, one can observe that AuNPs had a higher response. This can be explained by differences in the concentration of particles in the initial solution and of dialyzed bubbles samples, and localization of particles in a bubble. Still, AuNPs saline solution’s response primarily in low frequencies was achieved in the phantom’s entire sample volume, which correlates with the examples in the literature. The same behavior was observed for ICG-containing samples (BSA-ICG and AuNPs solution, BSA-ICG-AuNPs MBs). 

Bubbles containing both AuNPs and photodynamic dyes (ICG, ZnPc) demonstrated a higher photoacoustic response during measurements than bubbles where the shell was stabilized by AuNPs only; the highest PA signal corresponded to the BSA-ICG-AuNPs MBs sample. Still, all bubbles-containing samples had comparable extinction properties at the wavelength of 532 nm used for photoacoustic imaging characterization (Appendix A) and the same concentration of bubbles in the sample. Thus, the frequency of 33 MHz was validated as the optimal for PA imaging modality and suggested for further characterization in US imaging.

Next, acoustic characterization revealed the possibility of using the obtained probes as US contrast agents. Microbubbles were tested at a frequency of 33 MHz. The saline solution was taken as a control sample. All microbubbles-containing samples revealed a significant acoustic response compared with saline solution only, which showed no acoustic response, as shown in Figure 9.

The bubble’s gas core provided an excellent acoustic response for US imaging; similar behavior was observed for MBs containing dyes, only in the shell (BSA-ZnPc MBs and BSA-ICG MBs) and for MBs stabilized with AuNPs/dyes structures (BSA-ZnPc-AuNPs MBs and BSA-ICG-AuNPs MBs), respectively.

Thus, BSA-ICG-AuNPs MBs and BSA-ZnPc-AuNPs MBs demonstrated the possibility of using trimodal (FL/PA/US) imaging; BSA-ICG-AuNPs MBs were the most efficient imaging probe described in this article. Bubbles’ shell modification with photodynamic dyes, such as ICG and ZnPc, has promising potential for use in sono-/photodynamic therapy for dye-containing probes (BSA-ICG-AuNPs MBs, BSA-ICG MBs, BSA-ZnPc-AuNPs MBs, BSA-ZnPc MBs), opening up the possibility of theranostics applications based on the use of microbubbles.

## 4. Conclusions

The preparation conditions for microbubbles used in the modified sonication method (which were chosen as optimal during the work) allowed us to produce stable monodisperse microbubbles with an average size of 1.5 ± 0.3 μm. Additional stabilization of bubbles’ shell consisted of BSA with AuNPs/ICG, and AuNPs/ZnPc hybrid structures, which revealed an improvement in concentration, stability and the mean size of microbubbles This also opened the possibility of PA/FL imaging applications, in addition to the US imaging modality provided by the gas core, and promising applications in photodynamic/sonodynamic therapy to achieve combinations for theranostics. Due to the choice of components (BSA-coated gold nanoparticles and clinically available photodynamic dyes) and methods of preparation (all components were dissolved and dialyzed in saline solutions), biocompatibility, and further transition into preclinical (in vivo experiments) practice can be achieved. It was shown that even at low concentrations, AuNPs demonstrated increase stability and monodispersity of the produced probes. Cross-linking between photodynamic dyes (ZnPc, ICG) with a protein (BSA) was confirmed by the mass spectrometry measurements. The introduction of the dye into the system reduced the surface tension of the solution, leading to an increase in the concentration and stability of the bubbles. During imaging applications characterization, the use of bubbles stabilized both with ICG/ZnPc dye and AuNPs was observed—these probes demonstrated effective response during fluorescent tomography measurements, raster-scanning optoacoustic mesoscopy measurements, and ultrasound characterization.

The most relevant protein-dye pair for further applications in medicine was BSA-ICG. Subsequently, bubbles were obtained on the basis of such a multifunctional additive. Since ICG is widely used in clinical practice and an FDA-approved agent, albumin is a biocompatible protein present in the human body, and ICG-containing bubbles (BSA-ICG MBs and BSA-ICG-AuNPs MBs) provide the greatest stability, a higher concentration of bubbles and optimal signal enhancement for FL/PA/US multimodal imaging.

## Figures and Tables

**Figure 1 nanomaterials-11-00415-f001:**
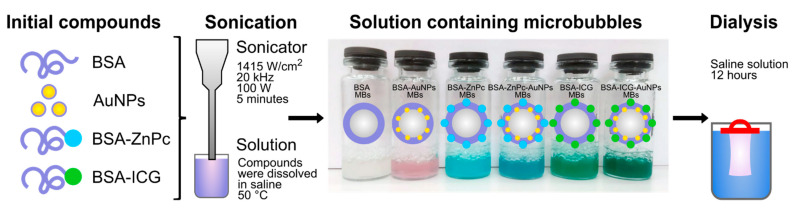
Scheme of microbubbles production by sonication method: bovine serum albumin (BSA) was used as the key component for the shell preparation, such compounds as BSA-coated gold nanoparticles (AuNPs) and conjugates of BSA with photodynamic dyes, zinc phthalocyanine (ZnPc) and indocyanine green (ICG), (BSA-ZnPc and BSA-ICG) were used for further bubbles’ shell stabilization. All compounds were dissolved in saline and then obtained solutions were sonicated for 5 min to obtain microbubbles. Resulting microbubbles with BSA shell (BSA MBs), shell stabilized with AuNPs only (BSA-AuNPs MBs), shell stabilized with BSA-ZnPc/BSA-ICG conjugates only (BSA-ZnPc MBs, BSA-ICG MBs, respectively), and shell stabilized both with conjugates and AuNPs (BSA-ZnPc-AuNPs MBs, BSA-ICG-AuNPs MBs) were then dialyzed in saline to remove free components from the solution.

**Figure 2 nanomaterials-11-00415-f002:**
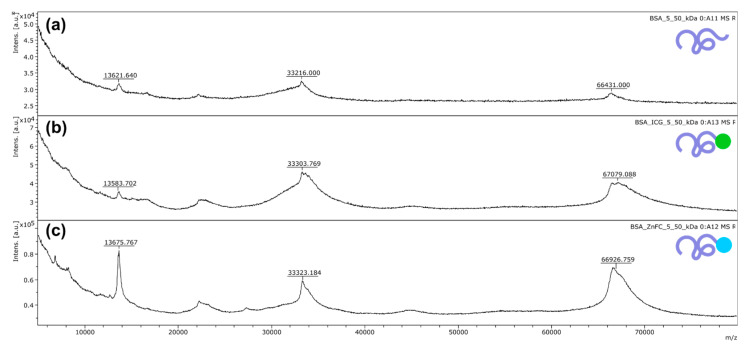
Mass spectrometry measurements for (**a**) BSA aqueous solution, (**b**) BSA-ICG conjugate aqueous solution, and (**c**) BSA-ZnPc conjugate aqueous solution were carried out to confirm the covalent binding of BSA-ICG and BSA-ZnPc conjugates.

**Figure 3 nanomaterials-11-00415-f003:**
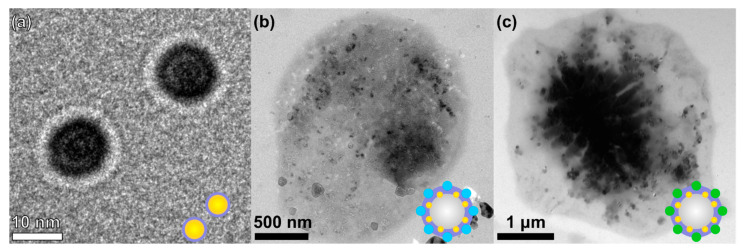
TEM images of (**a**) AuNPs coated with BSA shell, (**b**) dried BSA-ZnPc-AuNPs MBs sample, and (**c**) BSA-ICG-AuNPs MBs.

**Figure 4 nanomaterials-11-00415-f004:**
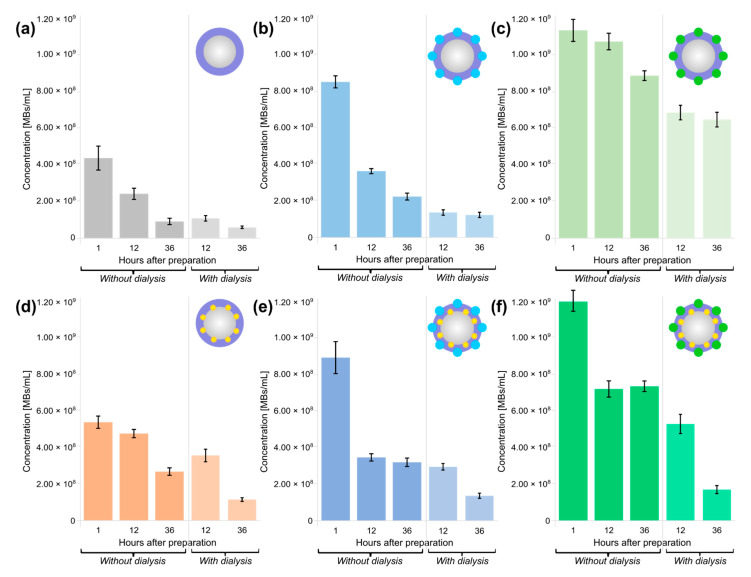
Bubbles concentrations and stability during storage time for probes without dialysis at 1, 12, and 36 h after prep-aration, and for dialyzed probes at 12 and 36 h after preparation: (**a**) microbubbles with the BSA shell only (BSA MBs), (**b**) microbubbles functionalized with ZnPc (BSA-ZnPc MBs), (**c**) microbubbles functionalized with ICG (BSA-ICG MBs), (**d**) microbubbles stabilized with AuNPs (BSA-AuNPs MBs), (**e**) microbubbles stabilized with AuNPs and ZnPc (BSA-ZnPc-AuNPs MBs), (**f**) microbubbles stabilized with AuNPs and ZnPc (BSA-ZnPc-AuNPs MBs). All samples were stored at 4 °C.

**Figure 5 nanomaterials-11-00415-f005:**
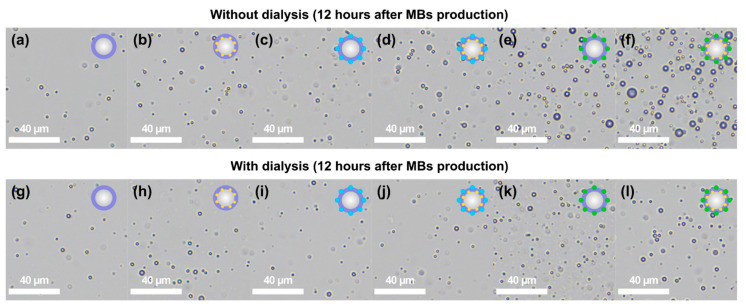
Optical microscopy (OM) images of air-filled bubbles 12 h after the preparation: (**a**) BSA MBs, (**b**) BSA-AuNPs MBs, (**c**) BSA-ZnPc MBs, (**d**) BSA-ZnPc-AuNPs MBs, (**e**) BSA-ICG MBs, (**f**) BSA-ICG-AuNPs MBs without dialysis, and (**g**) BSA MBs, (**h**) BSA-AuNPs MBs, (**i**) BSA-ZnPc MBs, (**j**) BSA-ZnPc-AuNPs MBs, (**k**) BSA-ICG MBs, (**l**) BSA-ICG-AuNPs MBs with dialysis.

**Figure 6 nanomaterials-11-00415-f006:**
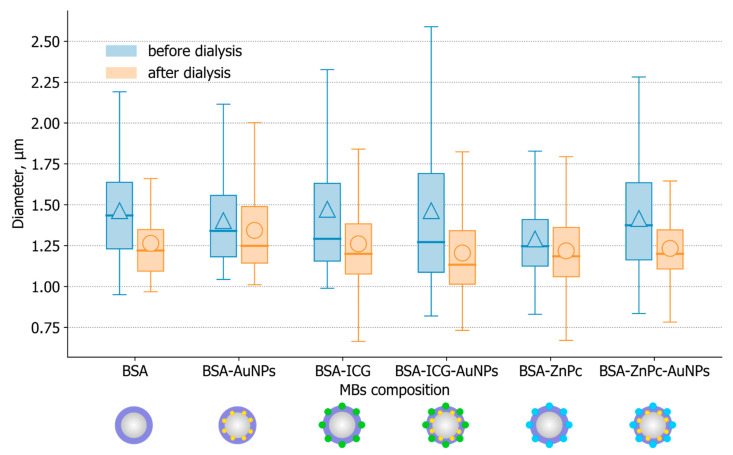
Mean size measurements for obtained microbubbles: measurements were made for samples treated with and without dialysis 12 h after probes’ preparation. All samples were stored at 4 °C.

**Figure 7 nanomaterials-11-00415-f007:**
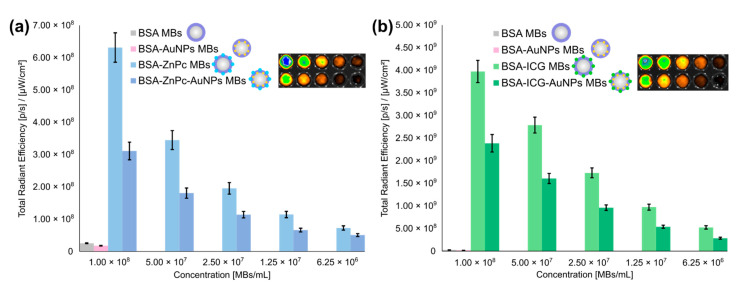
Fluorescence imaging of probes: comparison of total radiant efficiency dependencies on the concentration of MBs for (**a**) BSA MBs, BSA-AuNPs MBs, BSA-ZnPc MBs and BSA-ZnPc-AuNPs MBs at excitation/emission pair of 675/720 nm, (**b**) BSA MBs, BSA-AuNPs MBs, BSA-ICG MBs and BSA-ICG-AuNPs MBs at excitation/emission pair of 745/840 nm. Inlets on each figure show fluorescence imaging of a plate with BSA-ZnPc MBs, BSA-ZnPc-AuNPs MBs, and BSA-ICG MBs, BSA-ICG-AuNPs MBs solutions, respectively, in concentrations used in the plot.

**Figure 8 nanomaterials-11-00415-f008:**
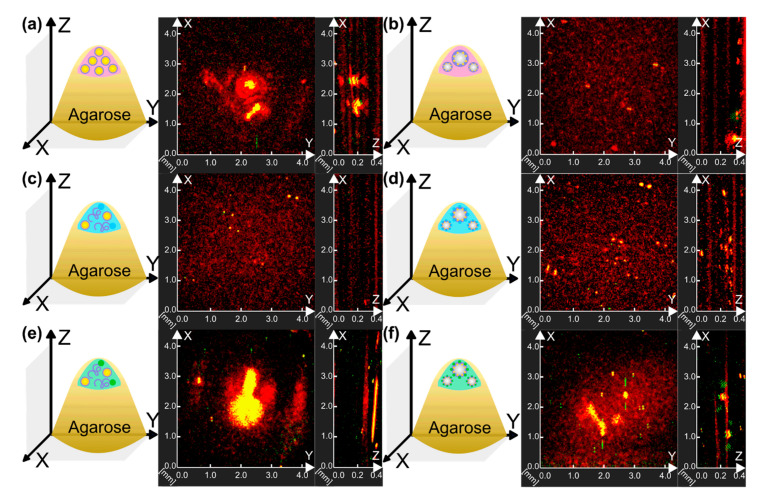
Schematic representation of the agarose phantoms and raster-scanning optoacoustic mesoscopy measurements of (**a**) AuNPs solution, (**b**) BSA-AuNPs MBs, (**c**) BSA-ZnPc and AuNPs solution, (**d**) BSA-ZnPc-AuNPs MBs, (**e**) BSA-ICG and AuNPs solution, (**f**) BSA-ICG-AuNPs MBs. Projections of measurements in X and Y, X and Z axes are presented, and the corresponding projections are marked in gray in the schematic representations for each measurement.

**Figure 9 nanomaterials-11-00415-f009:**
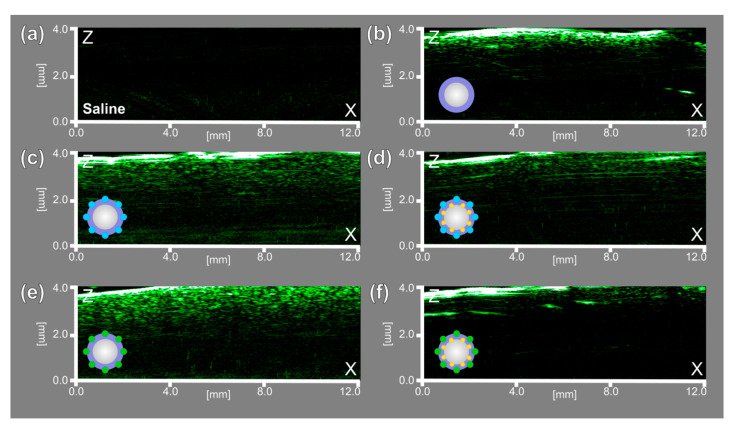
Ultrasound (US) imaging at a frequency of 33 MHz for (**a**) saline solution, (**b**) BSA MBs, (**c**) BSA-ZnPc MBs, (**d**) BSA-ZnPc-AuNPs MBs, (**e**) BSA-ICG MBs, (**f**) BSA-ICG-AuNPs MBs.

## Data Availability

Data is contained within the article and Appendix A.

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
