# Peer review of "Air-Filled Bubbles Stabilized by Gold Nanoparticle/Photodynamic Dye Hybrid Structures for Theranostics"

_nanomaterials, 2021, doi:10.3390/nano11020415_

Round 1

Reviewer 1 Report

This is an important work on development and characterization of novel air-filled stabilized microbubbles which can be used for multi-modal imaging and theranostics. The results obtained in the study should be useful for readers of the journal. These microbubbles may be used as contrast agents in optoacoustic, fluorescence, and ultrasound imaging as well as for photodynamic and photothermal therapy. The paper may be accepted for publication after a minor revision.

Please, address the following comments:

The abstract should be more informative. Please, include in the abstract the actual results obtained in the study such as major results on the quantitative characterization of the microbubbles.

Minor mistakes in English and style should be corrected:

In the sentence “To ensure unimpeded movement through the circulatory system, such microbubbles’ the optimum diameter is less than 10 μm.” The second “the” should be deleted.

In the sentence “30 mg of bovine serum albumin is dissolved in 7 mL of phosphate buffer (pH=8), 775 μL of EDC (solution in PBS with a concentration of 1 mg / mL) is added to the resulting solution, stirred for 15 minutes, then a solution of N-hydroxysuccinimide 1.05 mL (solution in PBS 1 mg/ml), mix for another 15 min, and add 1 ml of holosens (1 mg/mL).” “ml” should be replaced with “mL”. Please, choose “min” or “minutes” and use it in the manuscript. Please, correct other similar mistakes in the whole manuscript.

Please, use scientific notations in the sentence “All samples were diluted in saline with concentrations of 100000000, 50000000, 25000000, 12500000, 6250000 bubbles/mL.”

In the sentence “. For fluorescence tomography measurements, each sample was diluted in saline and added in a 96 well plate in the same manner as for extinction spectra measurements.” the period at the beginning of the sentence should be deleted.

Please, replace “pulse length” with “pulse duration” (it is a more established term) in the sentence “The samples were irradiated by a frequency-doubled flashlamp-pumped Nd:YAG laser (532 nm, pulse length - 2.5 ns; 200 μJ pulse-1; repetition rate - 1-2 kHz).” What does “200 μJ pulse-1” mean? If it is pulse energy, please, state so. Add “wavelength” before “532 nm”. What was the repletion rate in this study: 1 kHz or 2 kHz, or something in between?

The TEM results are provided in Fig 3, not in Fig 2. Please, correct it in this and other sentences: “Results of TEM provided in Figure 2a proved the shape of nanoparticles and the protein shell's presence with the size ~ 2 nm used for microbubble stabilization.”

Author Response

Answers to questions and comments of review reports on the Manuscript:
nanomaterials-1078001
Title: "Air-filled bubbles stabilized by gold nanoparticle/ photodynamic dye hybrid structures for theranostics”

Thank you for giving us the opportunity to submit a revised draft of our manuscript titled Air-filled bubbles stabilized by gold nanoparticle/ photodynamic dye hybrid structures for theranostics to Nanomaterials. We appreciate the time and effort that you and the reviewers have dedicated to providing your valuable feedback on our manuscript so that we can strengthen it prior to publication. We are grateful to the reviewers for their insightful comments, questions, advices for our paper. We have been able to incorporate changes to reflect most of the suggestions provided by the reviewers. The changes within the manuscript are marked in yellow in the edited version of the manuscript.

Here is a point-by-point response to the reviewers' comments and concerns.

Comments from Reviewer 1:

  • Comment 1: The abstract should be more informative. Please, include in the abstract the actual results obtained in the study such as major results on the quantitative characterization of the microbubbles.

Response: Thank you for pointing this out. The abstract was improved, and actual results obtained during characterization were described: “Microbubbles with an average size of 1.5±0.3 μm and concentration up to 1.2*109 microbubbles/mL are obtained and characterized. The introduction of the dye into the system reduced the solution's surface tension, leading to an increase in the concentration and stability of bubbles.” Such corrections were made in the manuscript.

  • Comment 2: In the sentence “To ensure unimpeded movement through the circulatory system, such microbubbles’ the optimum diameter is less than 10 μm.” The second “the” should be deleted.

Response: Thank you, the second “the” in this sentence was deleted in the manuscript.

  • Comment 3: In the sentence “30 mg of bovine serum albumin is dissolved in 7 mL of phosphate buffer (pH=8), 775 μL of EDC (solution in PBS with a concentration of 1 mg / mL) is added to the resulting solution, stirred for 15 minutes, then a solution of N-hydroxysuccinimide 1.05 mL (solution in PBS 1 mg/ml), mix for another 15 min, and add 1 ml of holosens (1 mg/mL).” “ml” should be replaced with “mL”. Please, choose “min” or “minutes” and use it in the manuscript. Please, correct other similar mistakes in the whole manuscript.

Response: Thank you so much, we change each “min” in the text for “minutes”, and each “ml” for “mL” as correct versions and pointed each correction in the text.

  • Comment 4: Please, use scientific notations in the sentence “All samples were diluted in saline with concentrations of 100000000, 50000000, 25000000, 12500000, 6250000 bubbles/mL.”

Response: Thank you for this observation, samples concentrations were rewritten as “1*108, 5*107, 2.5*107, 1.25*107, 6.25*106  bubbles/mL” in the manuscript.

  • Comment 5: In the sentence “. For fluorescence tomography measurements, each sample was diluted in saline and added in a 96 well plate in the same manner as for extinction spectra measurements.” the period at the beginning of the sentence should be deleted.

Response: We agree with this comment, thank you. The period at the beginning of the sentence and method description was deleted.

  • Comment 6: Please, replace “pulse length” with “pulse duration” (it is a more established term) in the sentence “The samples were irradiated by a frequency-doubled flashlamp-pumped Nd:YAG laser (532 nm, pulse length - 2.5 ns; 200 μJ pulse-1; repetition rate - 1-2 kHz).” What does “200 μJ pulse-1” mean? If it is pulse energy, please, state so. Add “wavelength” before “532 nm”. What was the repletion rate in this study: 1 kHz or 2 kHz, or something in between?

Response: Thank you. We replaced this sentence with the following changes: “The samples were irradiated by a frequency-doubled flashlamp-pumped Nd:YAG laser (wavelength - 532 nm, pulse duration - 2.5 ns; pulse energy - 200 μJ; repetition rate - 1 kHz).” The repletion rate 1 kHz was used from the options of 0.5, 1 or 2 kHz.

  • Comment 7: The TEM results are provided in Fig 3, not in Fig 2. Please, correct it in this and other sentences: “Results of TEM provided in Figure 2a proved the shape of nanoparticles and the protein shell's presence with the size ~ 2 nm used for microbubble stabilization.”
    Response: Thank you very much for your comment. We made the following changes in the text: correction links for Figure 3 were made and corrected description of Figure 3 “TEM images of a) AuNPs coated with BSA shell, b) dried BSA-ZnPc-AuNPs MBs sample, and c) BSA-ICG-AuNPs MBs.” Is provided.

Sincerely, on behalf of all coauthors,

Prof. Dmitry Gorin,

Skolkovo Institute of Science and Technology

Reviewer 2 Report

In this article, a range of micro bubbles for multimodal imaging combined to theranostics is prepared, characterised and tested for their stability and imaging properties as a function of the shell architecture, in particular the presence of BSA-coated gold nanoparticles, together with photodynamic dyes. 

This study is timely and uses a fair selection of complementary characterisation techniques to support the conclusions. The state of the art is extensively discussed, and graphical content is of a good quality with lovely colourful schemes. Nice work.

With respect to experiments, the reproducibility of the bubble preparation should be further ormolu convincingly discussed. Here follows more specific comments:

Turkevich synthesis was used, and not seeded growth of e.g. gold nanorods, therefore you can remove the CTAB, NaBH4, and AgNO3 from your list of chemicals. Also these would cause biocompatibility issues if not fully removed. On the other hand, citrate or citric acid is missing from the list.

The close-up gold NP image seems a little odd. Was the microscope tuned to limit aberrations and the image properly focused? The contour of the NP should be sharper, the contrast of the inside more homogeneous, and the protein corona, if not enhanced, should appear very faint. Here the halo surrounding the particles is more likely due to a very strong spherical aberration, and no conclusion can be drawn. If no higher quality images are available, perhaps you should use a less magnified image to support the size/morphology of the gold NPs, and confirm the protein corona presence with another technique/ protein quantification.

Since DLS equipment is quoted in the materials and methods section, DLS and zeta potential results for the different gold nanoparticles synthesis steps would be welcome as an additional supporting material, together with the extinction spectra from the gold nanoparticles used in bubble preparation.

Is the increase in the number of bubbles mentioned in the presence of Au NPs at t=1h statistically significant? The values in the presence and absence of gold look too close to conclude to a significant effect. Significance should be discussed in the text.

Bubble counting was performed 5 times per point, but were these 5 measurements done on separate bubble preparations, or a single one? Repeatability of the bubble preparation should be discussed in more detail, as variability is likely to be large.

Line 415: “good” stability etc. - zeta potential results should be given in SI, and some numbers given in the text.

Lines 441-442: Criteria for contrast agents - please add a citation.

Line 510 and quite a few other instances in the rest of the text: “Bubbles contained both AuNPs and (…)” the correct formulation is “Bubbles containing both AuNPs and (…)”. On the other hand, “bubbles-contained probes” is correct if you mean the fluorescent probes (molecules).

Author Response

Answers to questions and comments of review reports on the Manuscript:
nanomaterials-1078001
Title: "Air-filled bubbles stabilized by gold nanoparticle/ photodynamic dye hybrid structures for theranostics”

Thank you for giving us the opportunity to submit a revised draft of our manuscript titled Air-filled bubbles stabilized by gold nanoparticle/ photodynamic dye hybrid structures for theranostics to Nanomaterials. We appreciate the time and effort that you and the reviewers have dedicated to providing your valuable feedback on our manuscript so that we can strengthen it prior to publication. We are grateful to the reviewers for their insightful comments, questions, advices for our paper. We have been able to incorporate changes to reflect most of the suggestions provided by the reviewers. The changes within the manuscript are marked in yellow in the edited version of the manuscript.

Here is a point-by-point response to the reviewers' comments and concerns.

Comments from Reviewer 2:

  • Comment 1: Turkevich synthesis was used, and not seeded growth of e.g. gold nanorods, therefore you can remove the CTAB, NaBH4, and AgNO3 from your list of chemicals. Also these would cause biocompatibility issues if not fully removed. On the other hand, citrate or citric acid is missing from the list.

Response: Thank you very much for pointing this out, indeed, initially several synthesis methods were used, in this regard, excess substances remained in the list of reagents. We corrected everything in accordance with the used method.

  • Comment 2: The close-up gold NP image seems a little odd. Was the microscope tuned to limit aberrations and the image properly focused? The contour of the NP should be sharper, the contrast of the inside more homogeneous, and the protein corona, if not enhanced, should appear very faint. Here the halo surrounding the particles is more likely due to a very strong spherical aberration, and no conclusion can be drawn. If no higher quality images are available, perhaps you should use a less magnified image to support the size/morphology of the gold NPs, and confirm the protein corona presence with another technique/ protein quantification.

Response: Thank you. Gold nanoparticles were filmed on a Titan Themis Z transmission electron microscope using an operator, a number of images were obtained, in all images, including less magnified images, the corona of the protein is observed, we attach several micrographs with different magnifications, perhaps they are more suitable to confirm our results. On the other hand, results were obtained on the change in size and surface charge by the method of dynamic light scattering, in more detail this is presented in the following answer. Also, it should be noted that from the point of view of theoretical chemistry, bovine serum albumin should always displace citrate from the surface of gold nanoparticles, since it contains sulfur. Gold forms covalent sulfur-gold bonds, which are much more stable compared to the electrostatic interaction of the gold surface with citrate ions.

  • Comment 3: Since DLS equipment is quoted in the materials and methods section, DLS and zeta potential results for the different gold nanoparticles synthesis steps would be welcome as an additional supporting material, together with the extinction spectra from the gold nanoparticles used in bubble preparation.

Response: Thank you. To confirm the coating of gold nanoparticles with a bovine serum albumin shell, we used the results of DLS and zeta potential, which varied in accordance with the data in the table. The table will be added to additional materials. The surface charge of citrate-stabilized gold nanoparticles is negative, while the surface charge of protein-coated gold nanoparticles, preliminarily transferred to water (pH 5.5), is close to zero, which corresponds to the presence of protein on the nanoparticle surface. There is also an increase in the dynamic size of the nanoparticle due to the protein shell. Thus, we added the Table S1 in SI.

Samples

Zeta-potential measurements (mV)

Mean size obtained with dynamic light scattering measurements (nm)

AuNPs

-49±0.9

15±2

AuNPs coated with BSA

-1±0.4

45±4

  • Comment 4: Is the increase in the number of bubbles mentioned in the presence of Au NPs at t=1h statistically significant? The values in the presence and absence of gold look too close to conclude to a significant effect. Significance should be discussed in the text.

Response: Thank you. This result was consistently observed for the entire series of reproducible experiments. Even in small amounts (50-60 nanoparticles per microbubble), gold nanoparticles contained in the shell of microbubbles increase stability, which is still a potentially important result for further research and the possibility of a more significant increase in microbubble concentrations.

  • Comment 5: Bubble counting was performed 5 times per point, but were these 5 measurements done on separate bubble preparations, or a single one? Repeatability of the bubble preparation should be discussed in more detail, as variability is likely to be large.

Response: Thank you. The bubble counting provided in the article was carried out on the same sample of bubbles, but on different portions. The reproducibility of the results obtained was assessed; all syntheses were performed at least 10 times. All results and methods are reproducible.

  • Comment 6: Line 415: “good” stability etc. - zeta potential results should be given in SI, and some numbers given in the text.

Response: Thank you so much. We added Table S2 with zeta-potential results for obtained microbubbles in Supporting Information and added results in the following sentence: “Zeta-potentials of all bubbles stabilized with nanoparticles- and/or dyes- hybrid structures revealed good stability with values of -5.3±0.9 mV for BSA-ZnPc-AuNPs MBs and -4.8±0.2 mV for BSA-ICG-AuNPs MBs samples, while the probe with BSA-only shell revealed moderate stability with value of -8.4±0.6 mV for BSA MBs sample, as presented in Table S2.”

  • Comment 7: Lines 441-442: Criteria for contrast agents - please add a citation.

Response: Thank you for pointing this out, we added citations for two relevant review articles published in 2020 describing such criteria as size of 1-10 microns and concentration of 108-109 microbubbles/mL: Stride, E.; Segers, T.; Lajoinie, G.; Cherkaoui, S.; Bettinger, T.; Versluis, M.; Borden, M. Microbubble Agents: New Directions. Ultrasound Med. Biol. 2020, 46, 1326–1343, doi:10.1016/j.ultrasmedbio.2020.01.027. as reference [52] and Köse, G.; Darguzyte, M.; Kiessling, F. Molecular ultrasound imaging. Nanomaterials 2020, 10, 1–28, doi:10.3390/nano10101935. as reference [53].

  • Comment 8: Line 510 and quite a few other instances in the rest of the text: “Bubbles contained both AuNPs and (…)” the correct formulation is “Bubbles containing both AuNPs and (…)”. On the other hand, “bubbles-contained probes” is correct if you mean the fluorescent probes (molecules).

Response: Thank you for your comment. We changed the formulation for “Bubbles containing both…” in the text, and “bubbles-contained probes” for “bubbles-contained samples” in the text.

Sincerely, on behalf of all coauthors,

Prof. Dmitry Gorin,

Skolkovo Institute of Science and Technology
